

# Evolutionary morphology of the rabbit skull

Brian Kraatz[1] and Emma Sherratt[2]

[1] Department of Anatomy, Western University of Health Sciences, United States
[2] Department of Evolution, Ecology and Genetics, Research School of Biology, The Australian National University, Canberra, ACT, Australia

## ABSTRACT

The skull of leporids (rabbits and hares) is highly transformed, typified by pronounced arching of the dorsal skull and ventral flexion of the facial region (i.e., facial tilt). Previous studies show that locomotor behavior influences aspects of cranial shape in leporids, and here we use an extensive 3D geometric morphometrics dataset to further explore what influences leporid cranial diversity. Facial tilt angle, a trait that strongly correlates with locomotor mode, significantly predicts the cranial shape variation captured by the primary axis of cranial shape space, and describes a small proportion (13.2%) of overall cranial shape variation in the clade. However, locomotor mode does not correlate with overall cranial shape variation in the clade, because there are two district morphologies of generalist species, and saltators and cursorial species have similar morphologies. Cranial shape changes due to phyletic size change (evolutionary allometry) also describes a small proportion (12.5%) of cranial shape variation in the clade, but this is largely driven by the smallest living leporid, the pygmy rabbit (*Brachylagus idahoensis*). By integrating phylogenetic history with our geometric morphometric data, we show that the leporid cranium exhibits weak phylogenetic signal and substantial homoplasy. Though these results make it difficult to reconstruct what the 'ancestral' leporid skull looked like, the fossil records suggest that dorsal arching and facial tilt could have occurred before the origin of the crown group. Lastly, our study highlights the diversity of cranial variation in crown leporids, and highlights a need for additional phylogenetic work that includes stem (fossil) leporids and includes morphological data that captures the transformed morphology of rabbits and hares.

## INTRODUCTION

Though there exists a clear functional relationship between the vertebrate skeleton and locomotion, there are more limited examples of how the skull (cranium and mandible complex) may relate to movement. Strong associations between cranial form and locomotion are rare among vertebrates (*Wake, 1993*); however, the correlation between basicranial flexion and bipedal locomotion within our own lineage has been extensively studied (see *Lieberman, Pearson & Mowbray, 2000* for a thorough review). In a far less studied system, the morphological transformations of the leporid (rabbits and hares) cranium are in many ways similar to those of anthropoid basicranial flexion (*DuBrul, 1950*, but see *Moore &*

Corresponding author
Brian Kraatz, bpkraatz@mac.com

*Spence, 1969*; *Jeffery & Cox, 2010* for further discussion). Both hominid and leporid skulls represent conditions in which the basicranial and facial regions of the cranium flex ventrally relative to one another, where the basicranium is considered the flexor in hominids, and the facial region as the flexor in leporids. While these cranial transformations have been extensively explored as they relate to locomotion in hominins, and briefly within leporids (*White & Keller, 1984*; *Bramble, 1989*), given the similarity in cranial transformations between these groups, rabbits and hares represent an ideal system to further understand the relationship between cranial form and locomotor function.

In a previous study, we described the ventral flexion of the cranial face in leporids as *facial tilt* (*Kraatz et al., 2015*). Leporids exhibit pronounced dorsal arching of the cranial roof as the facial region reflects ventrally relative to the basicranium (Fig. 1). In their radiographic study, *Vidal, Graf & Berthoz (1986)* demonstrate the facial cranium of *Oryctolagus cuniculus* is tilted ventrally relative to the basicranium in resting position, which was also discussed by *De Beer (1947)*. Though this condition is previously described qualitatively with regard to *Oryctolagus* (*Thompson, 1942*; *DuBrul, 1950*; *De Beer, 1947*), we used angular measurements (e.g., Fig. 1) to demonstrate that the degree of facial tilt strongly varies among a wide range of living leporid species (*Kraatz et al., 2015*). Most strikingly, our previous study also showed that the degree to which leporid faces tilt ventrally is strongly correlated to locomotor style. Leporid species with skulls that have limited facial tilt (high angles) are more likely to exhibit generalist locomotory modes (less hopping, at slower speeds) and that those with pronounced facial tilt (low angles) are more cursorial (high speed hopping) (*Kraatz et al., 2015*).

Measuring ventral flexion as a simple facial tilt angle may show strong predictive value because it records relative changes in position between distinct regions of the cranium (i.e., splanchnocranium and neurocranium; *Kraatz et al., 2015*). However, such a simple measurement likely oversimplifies the complex shape changes and structural rearrangements of the cranium related to facial tilt. Therefore, a robust, geometric morphometric approach is needed to characterize the complex geometry (shape) of the cranium and understand how overall cranial shape impacts upon morphological disparity and relates to function. In this paper, we build upon our previous finding that shows facial tilting appears to be a major trait driving leporid cranial functional morphology (*Kraatz et al., 2015*) and turn our attention to better understanding the highly-transformed nature of the leporid skull by taking a geometric morphometric approach to study the shape variation across leporids. Our aim is to capture the complexity of leporid cranial shape among species, examine its relationship to the simple angular measure of facial tilt and thus locomotion, and examine its evolutionary history more broadly. To this end, we have compiled a three-dimensional landmark-based data set digitized on micro-CT scanned crania with the goals to (1) expand our taxonomic coverage to better capture the diversity of crown leporids (Table 1), (2) conduct a robust (geometric morphometric) exploration of shape among leporids, (3) use molecular hypotheses of leporid evolution to examine how the highly transformed leporid skull evolved, particularly in relation to locomotion and also evolutionary size change (evolutionary allometry), and (4) quantitatively describe the

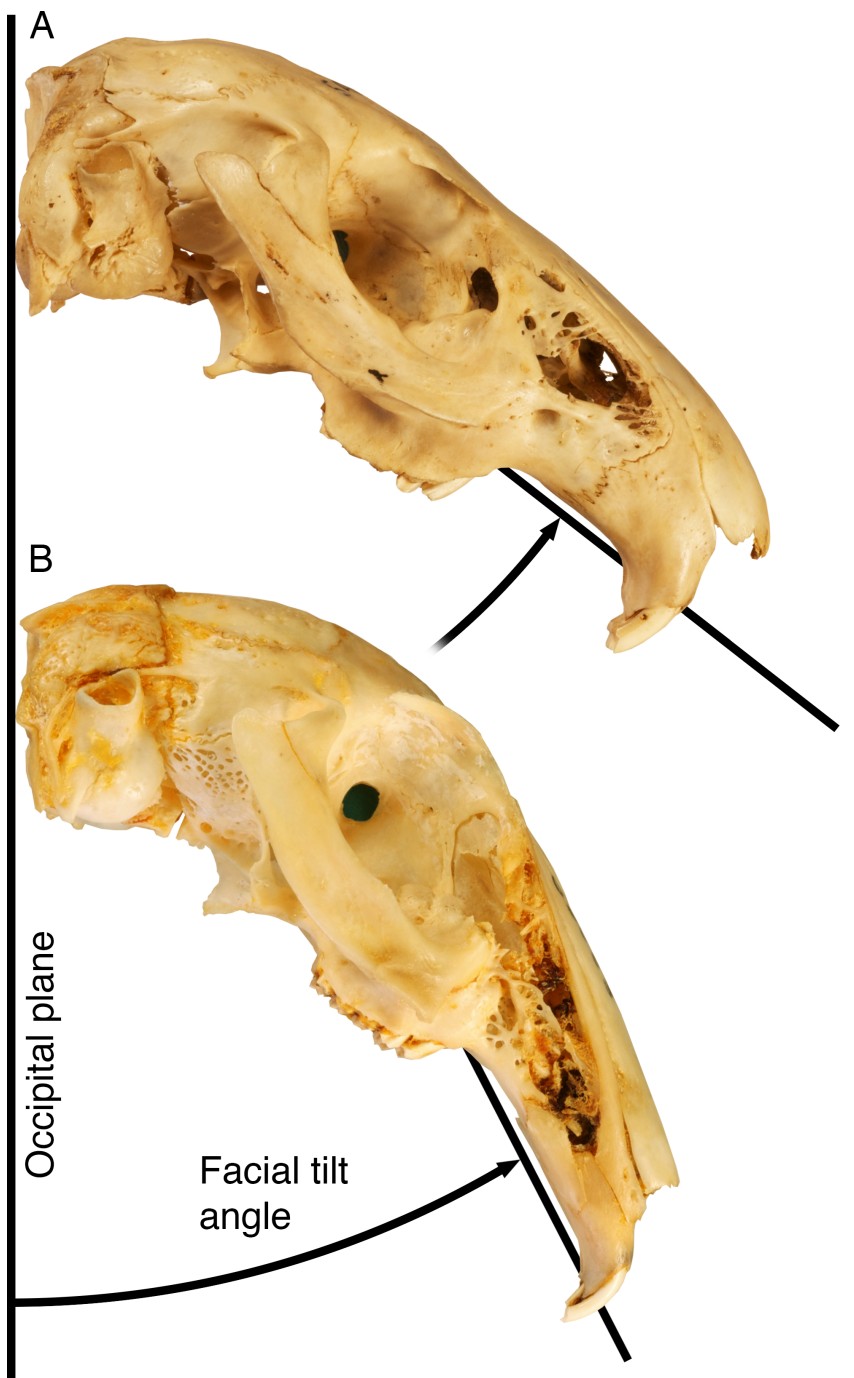

**Figure 1 Facial tilt in leporids.** The crania of *Caprolagus hispidus* (AMNH 54852, (A)) and *Pronolagus crassicaudatus* (AMNH 89033, (B)) are shown in right lateral view. Facial tilt (FT) is defined as the angle between the upper diastema and the occipital plane, where increased values indicated a skull orientation closer the horizontal plane, and calculated in this dataset was using the angle between landmarks 43, 1 and 7 (see Fig. 2). Modified from *Kraatz et al. (2015)*.
**Table 1  Leporid species included in this study.**

| Species | Locomotion type | Geographic range | n |
|---|---|---|---|
| *Romerolagus diazi* | Generalist | Central Mexico | 5 |
| *Bunolagus monticularis* | Saltatorial | South Africa | 4 |
| *Caprolagus hispidus* | Generalist | Himalayas | 1 |
| *Brachylagus idahoensis* | Generalist | NW United States | 12 |
| *Sylvilagus floridanus* | Saltatorial | Americas | 9 |
| *Sylvilagus palustris* | Generalist | SE United States | 9 |
| *Sylvilagus audobonii* | Saltatorial | Americas | 10 |
| *Sylvilagus aquaticus* | Saltatorial | United States | 9 |
| *Sylvilagus obscurus* | Saltatorial | Eastern United States | 10 |
| *Poelagus marjorita* | Saltatorial | Africa | 14 |
| *Pronolagus randensis* | Saltatorial | South Africa | 4 |
| *Pronolagus rupestris* | Saltatorial | South Africa | 14 |
| *Oryctolagus cuninculus* | Saltatorial | Global | 14 |
| *Nesolagus timminsi* | Saltatorial | Vietnam/Laos | 2 |
| *Pentalagus furnessi* | Generalist | Japan | 10 |
| *Lepus americanus* | Saltatorial | North America | 10 |
| *Lepus timidus* | Saltatorial | Old World, Palearctic | 12 |
| *Lepus capensis* | Cursorial | Africa, Arabia, Europe, Asia | 15 |
| *Lepus californicus* | Cursorial | SW North America | 10 |
| *Lepus saxatilis* | Cursorial | South Africa; Namibia | 10 |

complex morphological changes associated with facial tilt and assess the contribution of facial tilt to leporids cranial morphospace.

With these data we can explicitly test how cranial shape is driven by allometry, locomotor mode, and facial tilt angle (*sensu Kraatz et al., 2015*). If angular leporid facial tilt is a biologically-relevant trait that records changes in the facial region relative to the basicranium, we expect overall dorsal arching of the cranial roof to represent a close proxy for angular facial tilt, and therefore have a strong overall influence on cranial disparity. Our landmark scheme (Fig. 2; Table 2) was developed to capture this trait, although we have also measured angular facial tilt via our landmark data that closely matches previous angular measurements of facial tilt. We expect that skull roof dorsal arching and relative facial ventral flexion will load heavily on our first principal component axis, and be strongly correlated to angular facial tilt. If facial ventral flexion remains a dominant trait, and loads heavily on PC1 we also predict that both will be strongly associated with locomotor mode among leporids. Finally, given that a lack of facial tilt is a primitive condition for placental mammals and stem lagomorphs (*Asher et al., 2005*), we expect that our analyses estimating the evolutionary history of the leporids cranium will show that increased facial tilt is a derived condition within crown Leporidae.
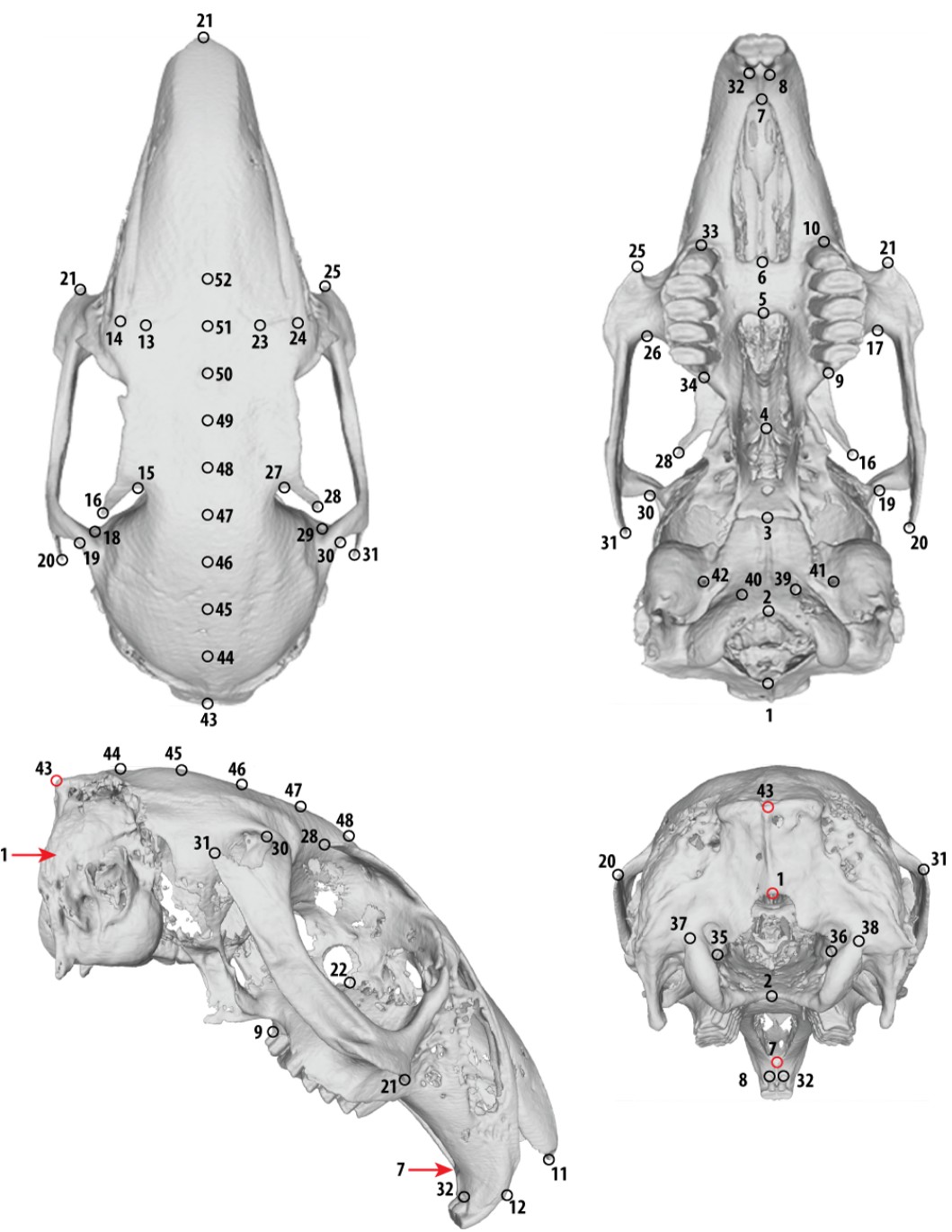

**Figure 2** **The 52 landmarks used in this study to characterize cranial shape.** Landmarks 43, 1 and 7 are marked in red in lateral and posterior views. See Table 2 for detailed explanation of landmarks.

## METHODS

### Samples and X-ray micro-CT

We sampled 184 leporid crania spanning 20 species (Table S1), including all 11 living genera of Leporidae. Species were chosen based on museum availability and to cover a breadth of the known diversity, while establishing sufficient coverage of the two most speciose

**Table 2** List landmarks used in geometric morphometrics analyses, illustrated in **Fig. 2**.

| Landmark number | Name (if applicable) | Description |
| --- | --- | --- |
| 1 | Opisthion | Midline point at the dorsal margin of the foramen magnum |
| 2 | Basion | Midline point at the ventral margin of the foramen magnum |
| 3 | | Anterior most point of basioccipital along midsagittal line |
| 4 | | Anterior most point of basisphenoid along midsagittal line |
| 5 | Staphilion (Alveolon) | Anterior most choanal opening on hard palate. In the leporid condition, this is condition changes from two parasagittal concavities (opening caudally), to singular concavity with age. In the presence of two concavities, the landmark is marked at the point that intersects a line drawn laterally between the anterior most point of both concavities, and the midsagittal line |
| 6 | | Midline posterior margin of incisive foramina; measured similarly to landmark 5 |
| 7 | | Anterior most point of incisive foramen at midsagittal line |
| 8 & 32 | | Posterior most point of alveoli of I3 |
| 9 & 34 | | Posterior margin of cheek tooth row |
| 10 & 33 | | Anterior margin of cheek tooth row |
| 11 | Rhinion | Anterior most nasal along midsagittal line |
| 12 | Nasospinale | Inferior most portion of nasal opening (premaxilla) |
| 13 & 23 | | Posterior most point of nasal on skull roof |
| 14 & 24 | | Posterior extent of premaxilla on skull roof |
| 15 & 27 | | Anterior most point of the posterior side of the root of supraorbital process |
| 16 & 28 | | Posterior extent of the supraorbital process |
| 17 & 26 | | Anterior most point of posterior margin of the maxillary root of the zygomatic arch within the orbital fossa |
| 18 & 29 | | Posterior most point of the anterior margin of the posterior root of the zygomatic arch within the orbital fossa |
| 19 & 30 | | Posterior most point of the posterior margin the squamosal root of the zygomatic arch |
| 20 & 31 | | Posterior extent of the posterior projecting jugal process of the zygomatic arch |
| 21 & 25 | | Anteroventral most point of masseteric spine |
| 22 | Chiasmatic sulcus | Point of contact between anterodorsal optic canals and presphenoid along midsagittal line as optic nerve emerges from braincase |
| 35 & 36 | | Lateral most point of the magnum foramen |
| 37 & 38 | | Dorsal most point of the articular surface of the occipital condyle |
| 39 & 40 | | Ventral most point of the articular surface of occipital condyle |
| 41 & 42 | | Medial most margin of hypoglossal foramen |
| 43 | External occipital protuberance | Posterior most point of the external occipital protuberance along the saggital plane |
| 44–41 | | Eight equidistantly placed landmarks placed along the saggital place between the external occipital protuberance (43) and Bregma (52) |
| 52 | Bregma | Posterior most point of suture between nasal bones |

genera, *Sylvilagus* and *Lepus*, and to best match the taxonomic coverage in the molecular phylogeny of *Matthee et al. (2004)*. To capture within-species variation, all efforts were made to include at least 10 crania of each species, although for some rare taxa this was not possible. Only adults were sampled, which were aged based on the degree of osteological fusion in the occipital complex (*Hoffmeister & Zimmerman, 1967*). The crania (dry skeletal

material without mandibles) were scanned using X-ray micro computed tomography (micro-CT) by J Morita using a Veraviewepacs 3D R100 system (typically, 70 kv, 3 mA) with voxel size ranging from 125–160 µm.

## Morphometric analyses

We characterized the shape of the 184 crania using landmark-based geometric morphometrics (*Bookstein, 1991*; *Mitteroecker & Gunz, 2009*). We thresholded each scan to obtain a 3D reconstructed model of the cranium and digitized 52 landmarks using the software Checkpoint (Stratovan Corporation, Davis, CA) (Fig. 2; Table 2); 44 of these are landmarks placed at homologous points on the cranium, over the left and right sides, and 8 are equally-spaced semilandmarks placed along the sagittal axis of the cranial roof to capture the curvature of the dorsal arch of the cranium. This curve of semilandmarks is homologous in all specimens and used to capture the changes in geometry of the cranial roof, an area which is highly variable among living leporids (*Kraatz et al., 2015*). Coordinate data for each specimen were exported as individual Morphologika files, which are entirely available as a combined, compressed Supplemental Information.

All of the statistical analyses were completed in *R* (*R Development Core Team, 2016*) using the package *geomorph* v.3.0.1 (*Adams & Otárola-Castillo, 2013*; *Adams, Collyer & Sherratt, 2016*). Of the 184 crania, 39 specimens were missing some landmarks, typically reflecting breaks in zygomatic arches, supraorbital process, or regions of the basicranium (denoted in the Morphologika files as 9999 9999 9999 coordinates). Missing landmarks were estimated with *geomorph* using a multivariate regression approach, where each missing landmark is predicted based on a regression among all other homologous landmarks of complete specimens within the data (*Gunz et al., 2009*). The missing landmarks for each specimen are summarized in Table S2. The landmark data were aligned using a generalized Procrustes superimposition (*Rohlf & Slice, 1990*), taking into account object symmetry, resulting in shape variables for the symmetric component of shape (*Klingenberg, Barluenga & Meyer, 2002*). During Procrustes superimposition the cranial roof semilandmarks were permitted to slide along their tangent directions in order to minimize Procrustes distance between specimens (*Gunz, Mitterocker & Bookstein, 2005*). The resulting symmetric shape data are used in the following analyses.

## Phylogenetic hypothesis

In order to examine cranial shape and factors influencing leporid cranial shape variation in a phylogenetic context, we used the phylogenetic relationships among species of Leporidae recently published by *Matthee et al. (2004)*. The original tree was constructed using seven genes (five nuclear and 2 mitochondrial) for 25 ingroup taxa. The tree was pruned to include only the 20 species studied here (Table 1) using Mesquite (*Maddison & Maddison, 2015*).

## Principal components analysis and phylomorphospace

The variation in cranial shape across all 184 specimens was first examined using Principal Components Analysis (PCA) of the symmetric shape data. Shape changes from the mean shape of the sample described by each PC axis were visualized using a surface warp approach (e.g., *Drake & Klingenberg, 2010*; *Sherratt et al., 2014*), which uses the thin-plate
spline (TPS) method (*Bookstein, 1989*). We took a triangular surface mesh (obtained from thresholding micro-CT for bone) of a specimen (*Lepus americanus*, LACM 70392, Table S1) close to the mean shape and warped it to the mean shape using TPS. Then we warped the mean shape mesh to the shapes represented by the minima and maxima of the first four PC axes.

A PCA of the species mean shapes was used to obtain a low-dimensional presentation of the leporid cranial morphospace, into which the phylogenetic tree was projected to estimate the evolutionary history of cranial shape change. We used maximum-likelihood ancestral state estimation to estimate the PC scores representing the internal nodes of the tree (using *fastAnc* in R package: *phytools*, *Revell, 2012*). The resulting phylomorphospace (*sensu Sidlauskas, 2008*) provides a visual representation of how the cranial shape of each species evolved in morphospace. As our data set does not include fossil taxa, the ability of our phylomorphospace analyses to reconstruct ancestral shape is limited. We discuss implications of the results with the fossil record in the discussion.

The amount of phylogenetic signal in cranial shape was estimated using the multivariate $K$ statistic (*Adams, 2014a*). The $K$ statistics provides a quantitative measure of the degree of homoplasy in the shape data and complements the species patterns visualized in phylomorphospace. A value of less than one implies that taxa resemble each other phenotypically less than expected under Brownian motion, and the test statistic is evaluated for significance using a permutation procedure, whereby the variables are randomized relative to the tree (*Blomberg, Garland Jr & Ives, 2003*; *Adams, 2014a*). We used 1,000 permutations.

## Comparative statistical analyses

Evolutionary allometry, the degree to which cranial shape variation among species is evolutionarily associated with size variation (*Klingenberg, 1996*), was examined using a phylogenetic generalized least squares (PGLS) approach and the pruned molecular tree of *Matthee et al. (2004)*. The size of each cranium was estimated using the centroid size of the 3D landmarks, which is calculated as the square root of the sum of squared distances of a set of landmarks from their centroid (*Dryden & Mardia, 1998*). We calculated species means of the symmetric shape data and centroid size for the 20 species. A Procrustes distance PGLS (D-PGLS) (*Adams, 2014b*) was done on the species means shape data and the natural log of centroid size. In this approach, significance testing is achieved by a permutation procedure, where the shape data are shuffled across the tips of the phylogeny (using the Randomized Residual Permutation Procedure; *Collyer, Sekora & Adams, 2015*), repeated 1,000 times, and estimates of statistical values are obtained and compared to the observed value.

We tested whether locomotor mode has an effect on cranial shape using a D-PGLS. Locomotory ability of each species was classified as in *Kraatz et al. (2015)*, except for that of *Romerolagus diazi*, which was previously considered to be saltatorial. Additional literature review reveals a generalist locomotor mode of *Romerolagus*, as described here, "It trots rather than hops as other rabbits do." (*Nowak, 1999*, pg. 1726). We colored species in the PC1 vs. PC2 morphospace by locomotor modes in order to visualize the morphospace occupation pattern relative to these categories.

We calculated facial tilt angle using our landmark data to best approximate how it was measured in our previous study (*Kraatz et al., 2015*): landmark 43 was the vertex, and we calculated the angle between two imagined lines whose endpoints are landmarks 1 and 7 (Fig. 2). First, we tested whether locomotor mode has an effect on facial tilt angle using a D-PGLS, to verify the new approach to calculating facial tilt angle and also given we have sampled more species than *Kraatz et al. (2015)*. Then, to understand how much facial tilt contributes to leporid morphospace, we examined (1) how much of the shape variation along PC1 of the species-mean PCA was attributed to facial tilt angle, and (2) how much of the total cranial shape variation was predicted by facial tilt angle, using a D-PGLS in each case. To describe the cranial shape changes associated with facial tilt, we performed a multivariate regression of the specimen symmetric shape data and facial tilt angle and summarized the shape variation predicted by facial tilt angle using the regression score (*Drake & Klingenberg, 2010*). Shape changes associated with a shift from the mean shape to the most ventrally-flexed (low angle) and least ventrally-flexed (high angle) were visualized using the surface warp approach, as described above for the PC axes.

## RESULTS

### Principal component analysis

Cranial shape disparity within leporids is driven by several key traits that can be summarized in the first four principal component analyses (Figs. S1 and S2), which account for 64.2% of explained shape variation in our overall data set; all subsequent PCs each account for 5% or less of the variation. Figure 3A illustrate biplots of combinations of the first three principal axes. The shape variation described by the first three axes is also summarized as warped crania surfaces (Fig. 4, Figs. S2 and S3), which are warps demonstrating the change from the mean shape to the minima and maxima of each PC axis (for PC4 see Figs. S2 and S3). The PCA suggests that three primary morphogroups (I–III) exist in leporid morphospace, and they are demarcated in Fig. 3A (PC1 vs. PC2).

PC1 describes changes associated with dorsal arching of the cranial roof (i.e., facial tilt) along the positive axis (Fig. 4, Figs. S2 and S3). Variation in cranial dorsal arching among our specimens is relatively continuous and strong throughout the entire data set. The positive axis of PC1 is also associated with facial or diastemata elongation and a widening of the proximal portion of the nasal bones. PC1 shows a reduction of bullae size towards the positive axis; however, our limited landmark coverage of that area suggests this shape change should be approached with caution. PC1 shows relatively continuous variation among our study group, but also distinguishes morphogroups I and II. *Brachylagus* is recognized as a distinct morphogroup due to separation along PC2 (see below). We note *S. palustris* as an outlier of morphogroup I, which trends toward the most negative space of morphogroup I near a member of morphogroup II (*Caprolagus*). Morphogroup II, which includes *Romerolagus, Pentalagus, Nesolagus,* and *Caprolagus,* clusters towards the negative portion of PC1. Within morphogroup I, all genera that include multiple species (i.e., *Lepus, Sylvilagus,* and *Pronolagus*) show species discrimination along PC1, and *Pronolagus* species are clearly separated by this axis.
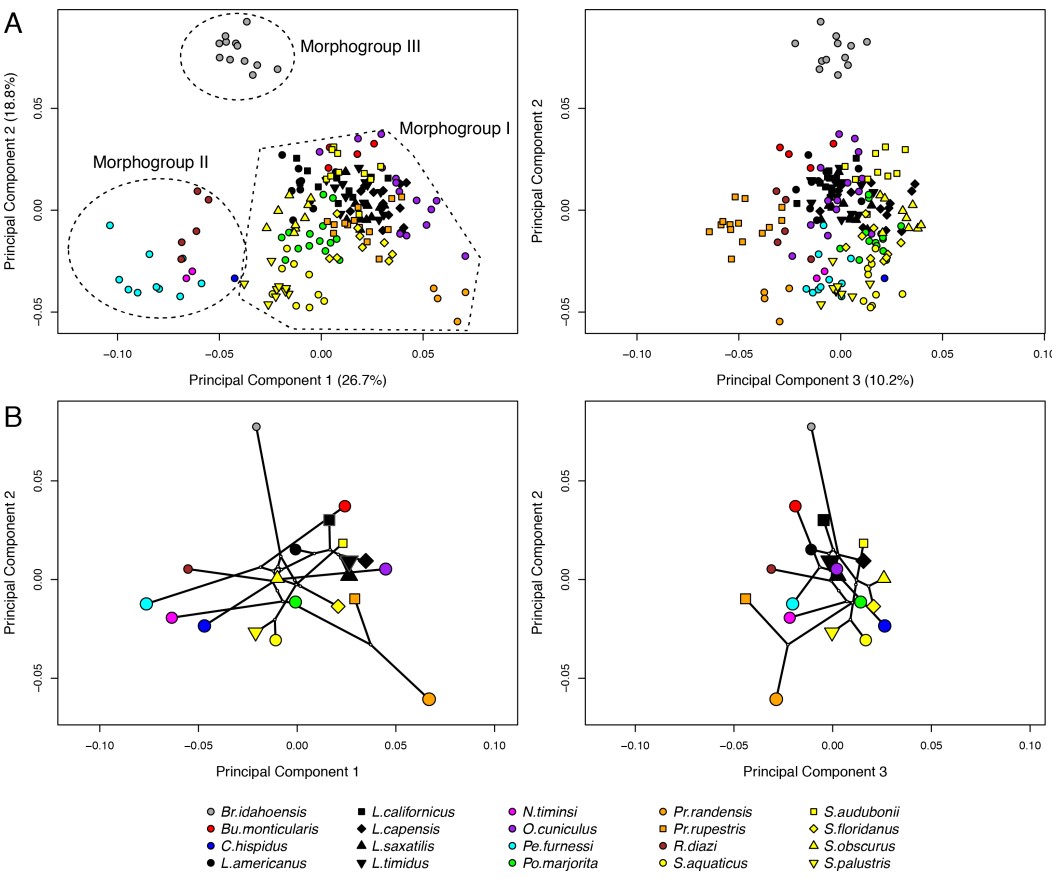

**Figure 3** **The leporid cranial morphospace and phylomorphospace.** The first three principal component (PC) axes from a PCA of 184 individual crania representing 20 species (A) The first three PC axes from a PCA of 20 species averages with the phylogeny of *Matthee et al. (2004)* projected into the shape space (B) Three distinct morphogroups can be distinguished. The legend provides a key to shapes and colors plotted in A and B. Shapes in B are scaled to average centroid size (1/80). Genus abbreviations in legend can be referred to full species names in Table 1.

Shape differences along PC2 are most strongly associated with relative proportions of the basicranial and the facial regions (Fig. 4, Figs. S2 and S3), where the basicranium is greatly enlarged relative to the facial region toward the positive end of PC2 as exemplified by *Brachylagus*. The orbit is also enlarged, the palate shortened, and the caudal ends of the zygoma flare laterally toward the positive portion of PC2. This axis strongly discriminates *Brachylagus* into a distinct morphogroup III. PC2 also discriminates among species within morphogroup I; *S. palustris*, *S. obscurus*, and *S. audobonii* show separation from a relatively more negative to more positive position along PC2, respectively. *Pronolagus* spp. are also separated along PC2, and both *Bunolagus* and *Poelagus* occupy distinct regions of morphospace within morphogroup I as delineated via PC2.

Shape changes associated with the negative axis of PC3 (Fig. 4 and Fig. S2) include diastemata elongation, narrowing of interorbital width (via frontal bones), and caudal retraction of rostral aspect of nasal bones. These shapes changes distinguish *Pronolagus* spp. from all other species in morphogroup I, and *Pronolagus rupestris* is a distinct outlier among

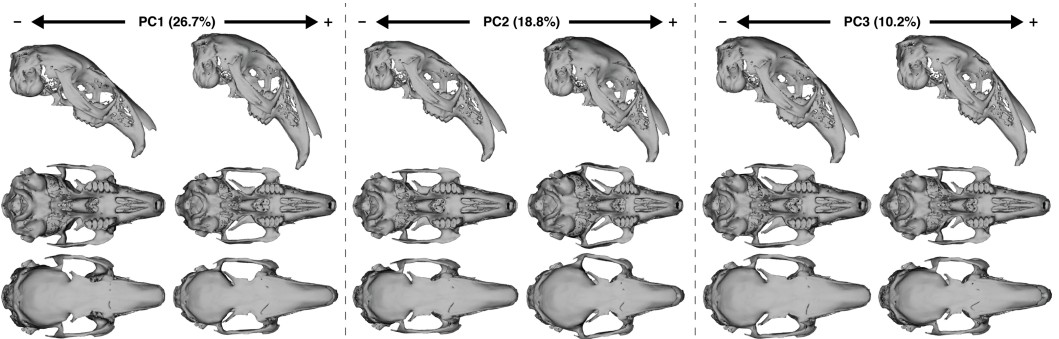

**Figure 4 The three main axes of shape variation in Leporid crania, as described by PCA of 184 specimens (Fig. 3A), visualized by warped crania surfaces.** Crania are shown in lateral, ventral and dorsal views, in two columns representing the shape change from the mean to the minimum value of the PC axis (denoted by a minus sign), and the shape change from the mean to the maximum value of the PC axis (denoted by a plus sign). Warping was done using thin-plate spline method (see 'Methods' for details). Lateral views were aligned along the occipital plane.

all species. PC3 also separates species within both *Sylvilagus* and *Lepus*; neither of which, however, overlap with *Pronolagus* within the morphospace of PC3. Along PC4 (Figs. S1–S3), the basicranium becomes narrower (as does the magnum foramen) and the supraoccipital processes extend caudally toward the positive aspect of that axis. This component distinguishes species within *Lepus* moderately well, which solely occupies the most positive aspect of PC4.

## Phylomorphospace

The specimen PCA (Fig. 3A) and phylomorphospace (Fig. 3B, and Fig. S4 for a 3D representation) reveal similar patterns of species occupation of morphospace and the main PC axes are congruent. Overall, the phylomorphospace shows that there is widespread homoplasy in crania shape; branches connecting sister taxa within morphospace often stretch far along PC axes. Much of this is driven by morphogroup II (relatively flat-skulled leporids), which is separated from the other morphogroups largely via PC1, yet most individuals within morphogroup II have immediate sister relationships outside of that group. In concordance with this pattern of homoplasy, we find there is no significant phylogenetic signal in cranial shape ($K = 0.70$, $P = 0.148$).

## Factors influencing cranial shape: facial tilt, evolutionary allometry, and locomotory mode

Across all 20 species, there is significant evolutionary allometry in cranial shape (D-PGLS, $F_{(1,18)} = 2.57$, $P = 0.005$); 12.5% of the variation in cranial shape is predicted by size. However, the pattern we found is clearly driven by the smallest species, the pygmy rabbit (*Brachylagus idahoensis*); excluding this species from the analysis revealed that only 7.7% of the shape variation was predicted by size, and this cranial shape-size relationship was not significant (D-PGLS, $F_{(1,17)} = 1.42$, $P = 0.157$). It is evident from the phylomorphospace, where points representing species means have been plotted and scaled to average centroid size (scaled by 0.02), that evolutionary allometry is not driving the main axes of shape

variation in leporids, since the different sized species are seemingly randomly distributed in morphospace (Fig. 3B).

Locomotory mode does not predict overall cranial shape in these 20 species (D-PGLS, $F_{(2,17)} = 2.20$, $P = 0.076$). Overlaying the three locomotor categories to PC1 versus PC2 plot (Fig. 5, left inset, from Fig. 3A) shows that generalist locomotors are strongly discriminated from both saltators and cursorial species along PC1. Generalists occupy only the negative portions of PC1, and show no overlap with cursorial taxa along that axis, yet there are two distinct groups of generalists, discriminated along PC2. All saltatorial and cursorial species are found within morphogroup I, excepting *Nesolagus* (morphogroup II). It is interesting to note that within *Sylvilagus*, *S. palustris* is the only species that exhibits a generalist form of locomotion and that species occupies the most negative space along PC1 for that genus, and the most negative space along PC1 for morphogroup I.

Facial tilt angle explains 21.6% of shape variation described by the first PC axis (PC1, D-PGLS, $F_{(1,18)} = 4.96$, $P = 0.03$), and 13.2% of the overall cranial shape variation among all 20 species (D-PGLS, $F_{(1,18)} = 2.69$, $P = 0.013$). We show the relationship between facial tilt and overall cranial shape in Fig. 6 using a multivariate regression; the regression score is a univariate summary of the highly multivariate shape changes associated with the independent variable, and thus can be thought of as an axis through morphospace that relates to variation in facial tilt angle. The shape changes most strongly related to changes in facial tilt include dorsal cranial arching and relative size changes between the basicranium and facial regions (Fig. 6). Incidentally, the regression score and PC1 are highly correlated (linear regression, $r^2 = 0.921$), signifying that the main axis of cranial variation in leporids (i.e., PC1) is strongly associated with dorsal arching and facial tilt.

Given locomotor mode does not predict cranial shape, but facial tilt does, it is noteworthy that we find these two factors to be themselves related. That is, in accordance with previous findings by *Kraatz et al. (2015)*, facial tilt angle is correlated with locomotor mode (D-PGLS, $F_{(2,17)} = 11.13$, $P = 0.003$), where lower facial tilt angle, meaning more pronounced cranial flexion, is found in cursorial species, and high angles are found in generalist species (Fig. 5 and Fig. S5).

## DISCUSSION

Our study strongly demonstrates that both dorsal arching and facial tilt have clear influences on the overall cranial shape of crown leporids, and that these influences are at least partially driven by ecological factors. The primary axis of leporid morphospace (PC1) characterizes a major portion of overall cranial shape variation among extant leporids (Fig. 4, 26.7%), and though several trait changes load strongly on that axis, this is the only PC axis that clearly highlights changes in both dorsal arching and facial tilt. We also show that facial tilt angle significantly explains one seventh of cranial shape among leporid species, and directly contributes to the shape differences along PC1.

Our analyses showed more mixed results with regard to the influence of allometric size changes on cranial shape. Although we found that there are significant differences in cranial shape associated with allometric size changes, this is largely driven by the smallest of living

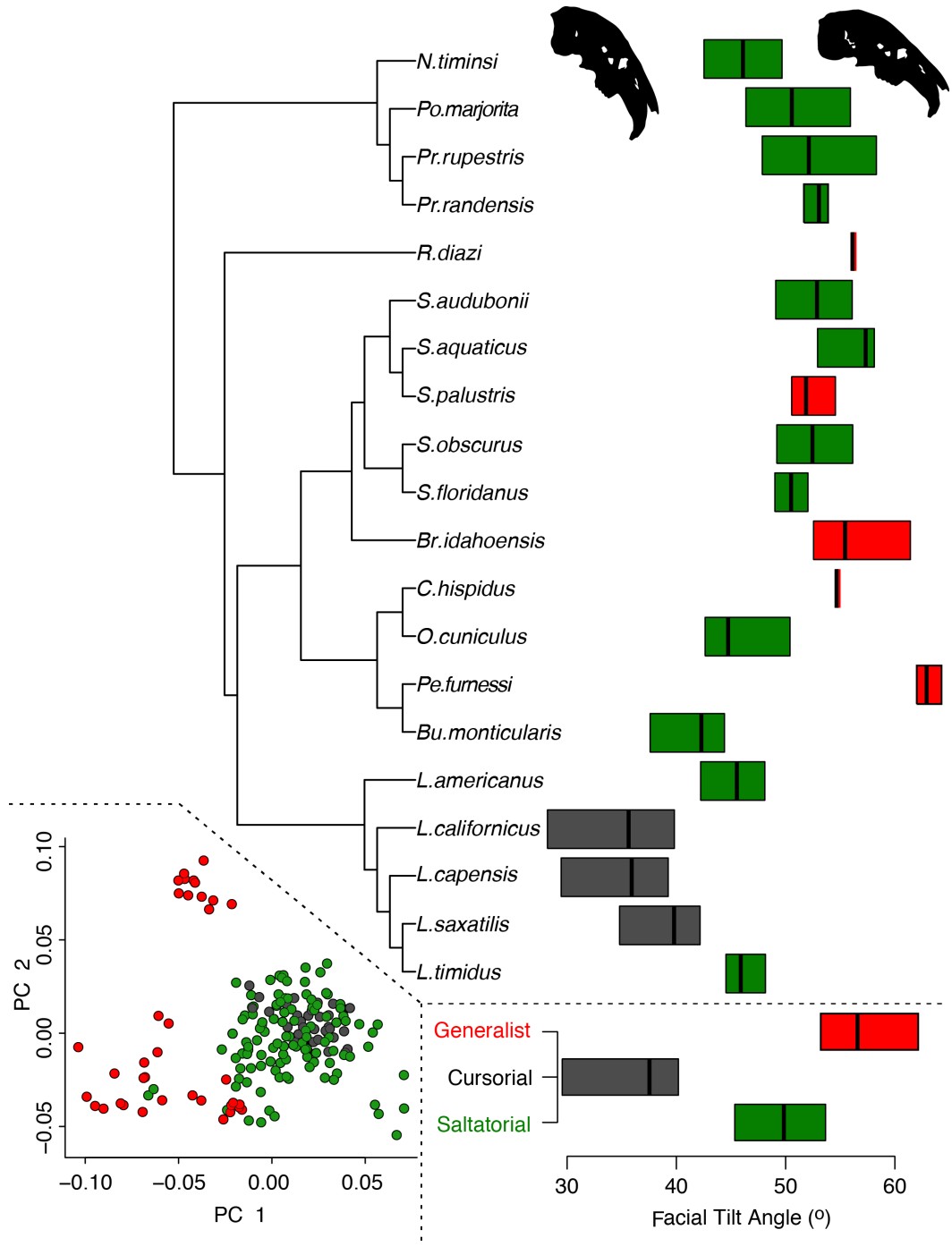

**Figure 5** **Summary of the relationship between locomotory mode and facial tilt, against the (*Matthee et al., 2004*) phylogeny.** Facial tilt angle (°) is plotted alongside the phylogeny. The boxes and midline represent the lower and upper quartiles (25% and 75%) and median of the facial tilt angle, colored by locomotory mode. A low angle represents a highly tilted cranium, as demonstrated by the silhouette crania. A summary of facial tilt angle for each locomotory modes. The cranial shape morphospace (as in Fig. 3A) is shown, colored by locomotory mode. See Figs. S5 and S6 for detail.

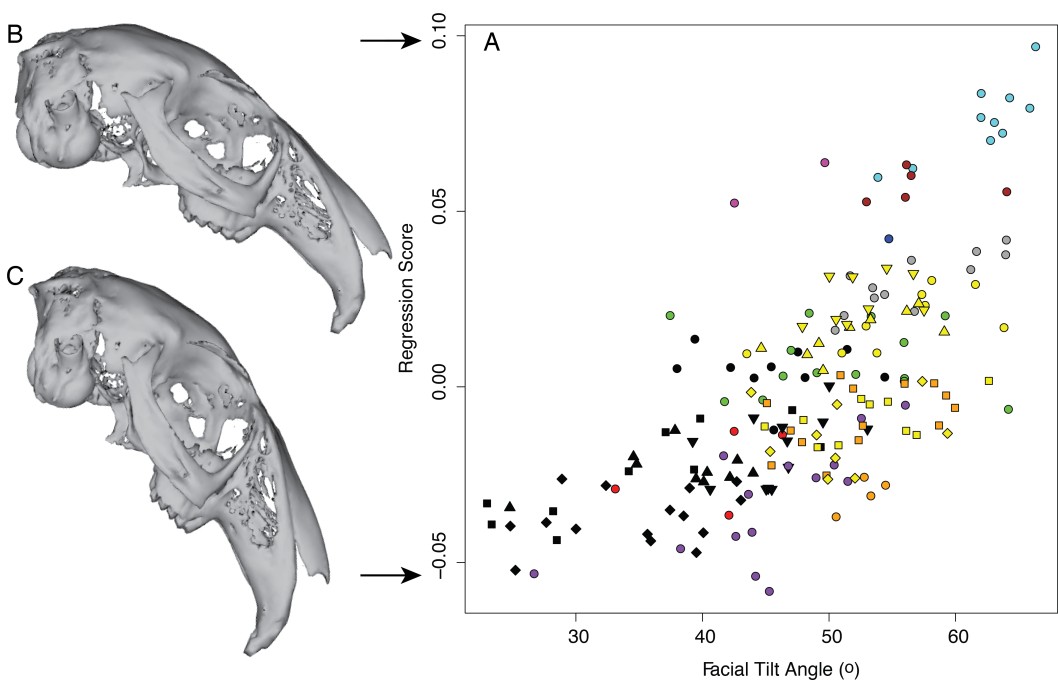

**Figure 6** **The relationship of cranial shape to facial tilt angle, as shown with a multivariate regression.**
(A) Cranial shape predicted by facial tilt angle is summarized as a regression score (sensu *Drake & Klingenberg, 2010*). Specimen points are denoted as in Fig. 3A. The warped crania (B & C) represent the predicted shape at the highest facial tilt angle, which corresponds to a positive regression score, and lowest facial tilt angle, which corresponds to a negative regression score.

leporids, the pygmy rabbit (*Brachylagus idahoensis*). The influences of the pygmy rabbit on allometric size changes are clearly a product of both its small size and unique overall cranial shape (Fig. 3). *Brachylagus* is isolated in morphospace, and strongly summarizes variation along PC2 due to its large basicranium region relative to its facial region. As demonstrated in many mammal lineages, as body size increases, the facial regions typically increase in size relative to basicrania (*Cardini & Polly, 2013*; *Cardini et al., 2015*). The short-faced *Brachylagus* is likely an excellent example of heterochronic changes within leporids that warrants further study.

Our study, along with *Kraatz et al. (2015)*, contributes to the growing body of literature that recognizes the need to quantitatively investigate the greatly underappreciated morphological disparity of the leporid skeleton. Certainly for taxonomic purposes linear measurements play an important role in delineating species (e.g., *Palacios et al., 2008*; *Pintur et al., 2014*). The laboratory model species *Oryctolagus cuniculus* has been extensively studied to understand pathologies of craniogenesis as they relate sutural synostosis (e.g., *Burrows et al., 1999*). *White & Keller (1984)* conducted a linear morphometric study to understand the ecomorphology of the North American lagomorph skull, and found evidence for three ecomorphs, which correspond to habit preferences, which we also recovered (and discuss further below). Our study particularly complements the 2D geometric morphometric analyses of postnatal growth changes and cranial disparity within leporids (*Ge et al., 2012*; *Zhang & Ge, 2014*; *Ge et al., 2015*), which showed postnatal

ontogenetic growth involves changes in the relative size of the basicranial and facial regions. Our findings align with those of *Ge et al. (2015)* who also found little phylogenetic signal within leporid morphospace, and high levels of homoplasy. Together, these studies and ours demonstrate to the broader audience the unappreciated complexity of the leporid cranium and highlight the need to investigate further what evolutionary and developmental factors have contributed to the morphological diversity of rabbits.

## Implications of leporid facial tilt angle and associated cranial shape changes

Locomotor mode influences significant aspects of cranial shape, namely the degree of facial tilt and dorsal arching. Here we show that overall cranial shape, particularly PC1 of leporid morphospace, is highly correlated to facial tilt angle. Our previous facial tilt angle measurement (*Kraatz et al., 2015*) was found to be significantly different between generalist locomotors and species that were saltators/cursors. Here we show that overall cranial shape, particularly PC1 of leporid morphospace, is highly correlated to facial tilt angle. Though this broad relationship between form and function is clear among leporid crania, the two most speciose genera, *Lepus* and *Sylvilagus*, each present important examples of this relationship at a more refined taxonomic level.

In their insightful study, *White & Keller (1984)* use a multivariate analysis of cranial linear measurements to show that cranial morphology describes three ecomorphs among North American lagomorphs (rabbits, hares, and pikas), irrespective of taxonomic relationships. Their study identifies *rock rabbit* (ochotonid, pikas), *cottontail*, and *jack rabbit* ecotypes, and points out that the snowshoe hare (*Lepus americanus*), while taxonomically a hare (*Lepus*), functions as a rabbit, and indeed, morphologically groups with *cottontails*. We identify a very similar pattern within our data set, where *L. americanus* groups in the most negative space along PC1 among all *Lepus* species and most species of morphogroup I (Fig. 3). As we have shown that PC1 is correlated with facial tilt, which itself predicts locomotor mode, there seems to be a clear change in cranial shape within *L. americanus* that is likely related to the fact that it ecologically resembles cottontails. The marsh rabbit (*Sylvilagus palustris*) represents a similar case within *Sylvilagus*, as *S. palustris* is known to have hind- and forelimbs that are roughly the same length and does not exhibit the typical hopping observed in cottontails (*Chapman & Ceballos, 1990*). As with *L. americanus*, *S. palustris* groups within the most negative space along PC1 among *Sylvilagus* species, and is more closely positioned to morphogroup II than any other species within morphogroup I (Fig. 3). The relative morphospace placement of *L. americanus* and *S. palustris* among closely related species suggests that there exists plasticity in cranial shape as it relates to function even within leporid genera, and strongly confirms the ecomorphs initially identified by *White & Keller (1984)*.

Though the facial tilt of leporids described in this study is not typical for the mammalian skull, this type of ventral flexion of the cranium is known from other mammalian groups. *DuBrul* (*1950*, plate 6), in his discussion of cranial arching in lagomorphs, contrasts parallel transformations in South American caviid rodents. He notes that the relatively flat-skulled, pika-like guinea pig (*Cavia*) is less facially tilted than the rabbit-like Mara (*Dolichotis*), and

given that guinea pigs are not cursorial and maras are, the correspondence between facial tilt and locomotion seems relevant in other closely related groups. *Spencer (1995)* showed that African bovids that preferentially fed on grasses have increased basicranial flexion; and in a 3D geometric morphometric study, *Merino, Milne & Vizcaíno (2005)* showed differences in basicranial flexion among cervids. *Drake (2011)* notes differences in basicranial flexion of dogs as compared to wolves, and *Wroe & Milne (2007)* show differences in basicranial flexion between marsupial and placental carnivores. Although these studies have shown similar examples of ventral flexion in crania, few have shown such a high degree of correlation with ecological variables, such *pronounced* ventral flexion, or such a significant influence on overall cranial disparity as shown here for leporids. Perhaps the only parallel of scale is that of anthropoids, which also show many of these features that are likely related to locomotor mode (*DuBrul, 1950*; *Lieberman, Pearson & Mowbray, 2000*). Basicranial flexion in hominids clearly relates to bipedal locomotion, but other factors, such as brain size increase (*Ross et al., 2004*) may also influence this trait. As we discussed previously (*Kraatz et al., 2015*), facial tilt in leporids also has the consequence of increasing orbital frontation that may allow for better visualization of the substrate during high-speed locomotion. While facial tilt is functionally predictive, it does not completely explain cranial shape, which is undoubtedly influenced by other important developmental, evolutionary, and functional factors. Though we have identified that one seventh of shape variation among leporid species is explained by facial tilt angle (and also a small but mostly not significant amount is due to evolutionary allometry), this leaves a large proportion unexplained. These important influences that drive evolutionary differences among leporids warrant further research.

## Inferring the evolutionary history of leporid cranial shape: insights from fossils

Understanding leporid cranial morphospace within the context of their evolutionary history illustrates that the morphological 'root' of crown leporids (*sensu* Leporinae, *Flynn et al., 2014*) is unclear. Our phylomorphospace plots (Fig. 3B and Fig. S4) show clearly that facial tilt is strongly homoplastic within crown leporids. Using fewer species from our data set, we also found the same homoplastic pattern using the molecular phylogeny of *Ge et al.* (*2013*; results not shown). *Ge et al. (2015)* conducted a recent 2D geometric study of lagomorph crania and found a similar pattern of homoplasy as it relates to cranial shape. Unfortunately, *Ge et al. (2015)* did not report lateral views of crania, so much of the facial tilt and dorsal arching patterns discussed here are not captured in that study.

Based on sister group relationships between each species in morphogroup II and individuals in morphogroup I, reduced facial tilt is dispersed throughout crown leporids, and therefore it has likely evolved multiple times (Figs. 3 and 5). The ancestral node to all species in this study is reconstructed very close to the origin of the phylomorphospace; this would produce a cranium that is subtly flexed with a facial tilt angle around 50° (average of the sample). But such an "average skull" is unsurprising given that ancestral state reconstruction is a method of weighted averaging, and the degree of homoplasy in cranial shape suggests that this method is not appropriate to provide a window into the ancestral

state of crown leporids. Instead, we must look to the fossil record to ask, did the ancestor of modern leporids have a flatter or more flexed facial region?

Leporidae and Ochotonidae (pikas) represent the two living families of lagomorphs, and recent fossil discoveries have revealed a detailed representation of stem lagomorphs. At deeper temporal levels, spanning to the base of the earliest stem lagomorphs, the polarity of facial tilt is one of increasing flexion of the facial region (Fig. 7). This stem lineage divided into the two modern families, but as discussed by *DuBrul (1950)*, ochotonids show little facial tilt. Facial tilt in the later stem lagomorphs and stem leporids as gleaned from the fossil record, may allow us to gain further insights into the origin of crown leporid cranial morphology.

The phylogenetic hypotheses of leporid evolution based on molecular evidence (*Matthee et al., 2004*; *Ge et al., 2013*) are not generally congruent with morphological evidence; the latter have been largely based on the morphology of the second and third premolars (*Flynn et al., 2014*, for a thorough discussion). Some recent studies have found more broad congruence between molecules and morphology at the level of stem lagomorphs (*Asher et al., 2005*; *Wible, 2007*), but there is not a clear picture of the evolutionary origin of crown leporids because there is no comprehensive phylogenetic study that includes crown leporids, stem leporids, and stem lagomorphs (see *Flynn et al., 2014*). Based on fossil teeth, it has been repeatedly suggested that both *Alilepus* and *Hypolagus* are sister to crown leporids and are near the origin of that clade (*Dawson, 1981*; *López-Martínez, 2008*; *Flynn et al., 2014*). Though much of the fossil record of leporids is based on partial maxillae, dentaries or isolated teeth, there are examples well-preserved skulls of both *Alilepus* and *Hypolagus* that illustrate skull shape in what is most likely the later stem-leporid lineage (Fig. 7). Both *Alilepus hibbardi* (*White, 1991*) and *Hypolagus* sp. (*Hibbard, 1969*) represent stem-leporid species that show more facial tilt than some extant lagomorphs, particularly those that fall into morphogroup II in our dataset (Fig. 3). *Hibbard (1969)* described the significant dorsal arching of the cranium of *Hypolagus* sp., and noted that it equals or exceeds that of any of the living leporids. In a morphometric study of the post-cranial skeleton of the European *H. beremendnesis*, *Fostowicz-Frelik (2007)* showed that the limb elongation of that species is comparable to that of highly cursorial extant leporid species. In short, various stem leporids had acquired a high degree of facial tilt that exceeds any measured in extant species.

## CONCLUSION

While a more comprehensive phylogenetic treatment that includes fossil taxa and better coverage of stem leporids is desperately needed, we can draw several important conclusions by comparing our results to what we know of leporid cranial morphology from the fossil record. We show here that facial tilt strongly influences the disparity of the leporid cranium, yet the appearance of that trait likely occurred well before the radiation of the crown group. This, however, does not clarify what the 'ancestral' crown leporid looked like. Though facial tilting predates the crown diversification, it's unclear how that trait varied among and within stem-leporid lineages. It remains entirely likely, for example, that

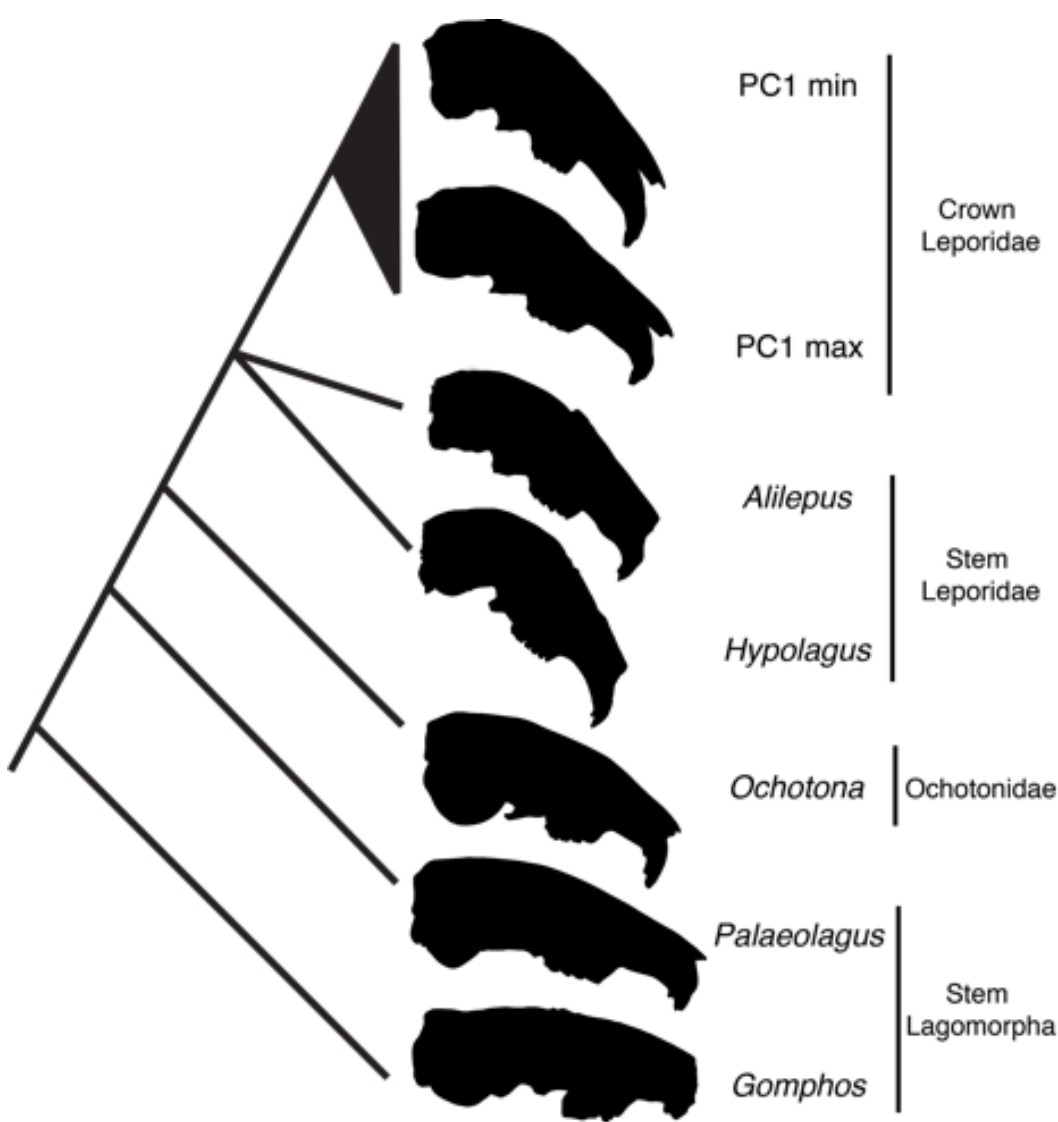

**Figure 7** Lateral views of fossil and extant species that represent the likely cranial macroevolutionary history of Lagomorpha, evolutionary relationships discussed in *Flynn et al. (2014)*. Silhouettes are from this study or modified from figures in studies cited. Crown Leporidae is represented by minimum and maximum surface warp transformations of PC1 (this study); stem-Leporidae is represented by *Alilepus hibbardi* (*White, 1991*) and *Hypolagus* sp. aff. *H. vetus* (*Hibbard, 1969*); Ochotonidae is represented by the only living genus, *Ochotona* (modified from drawing by Lily Li: see acknowledgements); stem lagomorphs are represented by the late appearing *Palaeolagus* (*Wood, 1940*) and early appearing *Gomphos* (MAE-14425). All crania are scaled to the same size

facial tilt and locomotor mode varied among *Hypolagus* or *Alilepus* species, as it does for living genera and species. Facial tilt is a derived trait in the context of stem lagomorphs; however, it seems clear that the lower degrees of facial tilt within various crown leporids has been independently acquired from a more 'facially tilted' stem leporid ancestor, and that facial tilt is an adaptively plastic trait within the crown group. Most importantly, a comprehensive, combined phylogenetic treatment of lagomorphs that more explicitly

covers stem leporids is badly needed. Such a treatment must thoroughly incorporate morphological data beyond teeth, as the evolutionary history of leporids is marked by remarkable and adaptively significant transformations of the skull.

## ACKNOWLEDGEMENTS

We would like to sincerely thank the many collections staff and museums that provided assistance and specimens for this study: Kawada Shin-ichiro, Department of Zoology, National Museum of Nature and Science (Tsukuba, Japan); Jim Dines, Department of Mammalogy, Natural History Museum of Los Angeles County (USA); Eileen Westwig and Neil Duncan, Department of Mammalogy, American Museum of Natural History (New York, USA); Esther Langan and Darrin Lunde, National Museum of Natural History, Smithsonian Institution (USA); Judith Chupasko, Department of Mammalogy, Museum of Comparative Zoology, Harvard University (USA). Nicolas Bumacod provided important help in developing the protocol to acquire geometric morphometric data. Assistance for micro-CT scanning was provided by Dr. Bruno Azevedo (University of Louisville) and Satareh Lavasani and Alex Lee in WU's College of Dental Medicine. Mathew Wedel provided thorough comments on an earlier version that greatly improved this manuscript. Lily Li is sincerely thanked for allowing us to use her wonderful original drawing of *Ochotona* in Fig. 7. I (BK) would especially like to thank Malcolm C. McKenna (deceased) for giving me the rabbits from his Mongolian fossil collections 15 years ago, when I asked for the insectivores, hyaenadontids, or the carnivores. He was right, people need to work on rabbits.

### Funding

BK received funding from the Western University of Health Sciences. The funders had no role in study design, data collection and analysis, decision to publish, or preparation of the manuscript.

### Grant Disclosures

The following grant information was disclosed by the authors:
Western University of Health Sciences.

### Competing Interests

Brian Kraatz is an Academic Editor for PeerJ.

### Author Contributions

- Brian Kraatz and Emma Sherratt conceived and designed the experiments, performed the experiments, analyzed the data, contributed reagents/materials/analysis tools, wrote the paper, prepared figures and/or tables, reviewed drafts of the paper.

## Data Availability

Kraatz, Brian (2016): Supplementary Materials_with media.pdf.

Figshare: https://dx.doi.org/10.6084/m9.figshare.3577956.v1.

## Supplemental Information

Supplemental information for this article can be found online at http://dx.doi.org/10.7717/peerj.2453#supplemental-information.

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
