# Peer review of "Evolutionary morphology of the rabbit skull"

_PeerJ, doi:10.7717/peerj.2453_

## Round 0.1 · original submission · Major Revisions

· Academic Editor

Major Revisions

Both reviewers have recommended major revisions, so this is what I am recommending overall. I was initially concerned on reading this manuscript on the overlap with your previous paper on leporids. I think attending to the reviewer comments will help make that differentiation - particularly with regard to pinning down some testable hypotheses. A more clear definition of facial tilt is also clearly needed.

I look forward to seeing your revisions.

Reviewer 1 ·

Basic reporting

This submission documents well the extant morphological variation of cranial shape in leporids. While the analysis of such a large dataset of µCT scanned crania via geometric morphometric methods deserves to be published, I have several suggestions to improve this manuscript.

My main concern is that no hypothesis is defined in the introduction. You make extensive use of geometric morphometric methods (with in my opinion you use well), but do not use these methods to test hypotheses. Rather, this work presents an exploration of cranial morphological variation, your aims remain vague. I would suggest that you define one or two evolutionary hypotheses in the introduction, and that you properly test them.

Furthermore, the abstract only poorly reflects the content of the manuscript: while you mention a lot the terms “facial tilt” in the abstract, I could find no figure explaining what facial tilt is and how it is measured. Neither could I find measurements of “facial tilt” throughout the manuscript. Also, while the abstract gives the feeling that you will (re)test for ecological correlates (and especially locomotion) of facial morphology, this is not the case.
Again in the abstract, you state: “we show that an increase in facial tilt likely occurred before the radiation of crown leporids. Within the crown group, a reduction in facial appears to be a homoplastic trait that has evolved multiple times”. I disagree: you did not test for this, neither do you discuss it. I could find no support for these 2 sentences in your manuscript. Your results rather support that facial tilt is a homoplastic trait. You write: “and recent fossil discoveries have revealed a detailed representation of stem-lagomorphs (see Kraatz et al, 2010, for discussion)”. => this reference lacks at the end of the manuscript. Is it “Evolutionary Patterns in the Dentition of Duplicidentata (Mammalia) and a Novel Trend in the Molarization of Premolars”? I could not see anything related to facial stem-lagomorph facial tilt in this publication.

So you should be careful when you state through your manuscript (in the abstract and for instance line 300) that “reduced facial tilt is dispersed throughout crown leporids, and therefore it has evolved multiple times” => well, the opposite may be true as well: increased facial tilt may have evolved several times.

My other concern is that manuscript was obviously written in a hurry. It contains far too many small mistakes of little importance. Please correct all the figures, tables, your list of references before submitting a new version to PeerJ. Also, improvements should be brought regarding the use of figure symbols (remark valid for figure 2, 4, 5, and S3), which do not relate to phylogeny and are at the moment confusing.

Experimental design

A central research question is lacking in this manuscript.

1) Your sample is large and well designed.
2) You are familiar with GM methods.

=> It would be easy that you define one or two evolutionary hypotheses in the introduction, and that you properly test them.

Validity of the findings

This work, in this state, is explorative. No hypothesis has been properly tested.

- Plus in my opinion the statement in the abstract "we show that an increase in facial tilt likely occurred before the radiation of crown leporids" is not supported by the results or properly discussed.

Additional comments

Important information are put in the supplementary data (tables S1, S2 and the phylogeny used throughout the manuscript at the end of figure S3), which would have their place in the core of the manuscript. Tables S1 and S2 need completion (see remarks below) and corrections. Regarding the supplementary data in general: why did you put a part on figshare, and another on PeerJ ?

Specific statements :
Line 34: a reduction in facial […]. The word “Tilt” is lacking.
Line 38: Are need that => are needed to?
Line 53: Fig. 1 does not relate to facial tilt. We have to look at Kraatz et al. to understand what you exactly mean by facial tilt. In the current state of this article, it cannot be read and understood on its own.
Line 60 : Kraatz et al. (2015) was => were ?
Lin 74 : Siparity ? Disparity?
Line 109, figure 1, table S2 : why did you digitize 10 landmarks and semilandmarks between the occipital bone and the bregma, and not a single one between the bregma (forgotten on table S2) and the rhinion (rhinion shoud be spelled with a “h” on table S2) ? Are there underlying hypotheses? Explain if you expected to capture the curvature mostly in the skull roof, and that the nasal region is not so important to capture cranial shape variation? Supplementary data figure S2: landmarks 43 to 52 definition are missing
< Species were also chosen to match the phylogenetic hypotheses used in this study (Matthee et al., 2004). => Some explanation would be welcome. This statement is unclear.


<117 “were missing some landmarks” : be precise : were missing between 1 and XXX landmarks

<115 and 120 : precise somewhere that the estimation of missing landmarks is done in Geomorph as well, and that the missing landmarks in the Morphologika files have 9999 9999 9999 coordinates.

<122 : explain why you have removed the asymmetrical component of shape variation. Did you expect to have a lot of skull shape asymmetry that would blur the other morphological signals ? Or did you follow recommendations given in some published work / geomorph’s user’s guide ?

<131 : be precise : which specimen did you use?
<149 (using fastAnc in phytools Revell 2012) => Precise that it’s a R package.
I am not a big fan of phylogmorphospaces when the sample only includes extant taxa. I would suggest here to add a few sentences with some emphasis on the difficulty to estimate ancestral shapes when no fossil is included in the sample (there could be some convergence …) . Of course this approach is useful in your case to show that the branches widely cross each other, and I appreciate this result.
< 232 : I would expect that you rather present in Fig. 5 the results of a linear discriminant analysis on the 3 locomotor categories, and display the percentage of specimens a posteriori correctly reallocated to their original locomotion category.
< 248 : while the relationship between facial tilt and locomotor mode is supported by the results of Kraatz et al. 2015, I would suggest you to consider to mitigate this statement here or to give a better support to this statement: you merely have shown a separation between generalists and others on PC1 + associated shape variation.
In the current state of the paper, facial tilt was not computed. No regression between facial tilt vs. PC1 scores (or other PCs) was performed.

< 263. The correlation between facial tilt and PC1 score was not computed. PC1 does not only relate to facial tilt changes. And almost three quarters of leporid shape variation is not explained by PC1. So I do not agree with the statement that facial tilt “is the most influential trait structuring the cranial morphospace”

< 300. Be precise: a list of specimens with the measurement of facial tilt + which specimens you consider to have a reduced facial tilt should appear somewhere to support this statement.

<540 (Figure 1) : which specimen is on the figure? + There are too MANY errors on this figure (landmark 21 appears 4 times etc…). + Table S2 is incomplete and contains also mistakes. Table S2 should be part of the manuscript, or at least on PeerJ, not on a foreign website.


<544 (Figure 2) : It would do no harm to repeat the % of variation held by PC1 and PC3 in the upper part of the figure. Please either attribute a single symbol (circle, square) to a single species species, or (if you consider that this phylogeny will still be valid in the future) use the 5 clusters that appear on the phylogeny (in figure S3) and use 1 symbol for each cluster.
So in the end, I would suggest that there would be either 20 different symbols used here (one for each species), or 5 symbols representing the 5 clusters I see on the phylogeny you use. The current use of the symbols is confusing. Please also scale symbols according to centroid size.
< This remarks regarding the use of confusing symbols is valid for figure 4, figure 5 and figure S3
<558 : a scale for centroid size would be welcome + the phylogeny shown on figure S3. What is the value of a phylomorphospace when no fossil data is included in the specimen sample?
<564 : scale dots to centroid size as well. A scale would be welcome.
<573 is represented by


<Supplementary table 1 should be made more readable and include locomotion mode. Also it should be part of the manuscript of at least not be hosted on a foreign website. I could also contain centroid size, facial tilt (you talk a lot about facial tilt, but never compute it anywhere).

<Supplementary figure 3, lower part: the phylogeny used in this study should appear somewhere in the manuscript, as it is the basis of many analyses performed in this manuscript.

Reviewer 2 ·

Basic reporting

There is significant overlap with Kaatz’s previous PeerJ paper (2015) – the main difference here being methodological. I think there is sufficient added value to justify the present paper but will defer to the judgement of the editor on this matter. The addition of endocranial landmarks would strengthen the paper (see below).

Experimental design

Since the crania were scanned with microCT I cannot understand why the analyses and findings are predicated on exocranial landmarks. Whilst external and internal cranial base morphologies can be correlated, there is some evidence, in primates at least, that the patterns differ (see review in Lieberman et al 2000). These missing data may well have proved informative, particularly with respect to allometry.

It is not clear how the lateral views in Figure 3 were standardised. Are these as they appear after TPS warping or have they been rotated in relation to the occipital plane as per Kaatz’s (2015)?

Were the data in Figure 5 size scaled as in Figure 4? Scaling appears has been done at the species level whereas the functional analysis was done with individuals. I suppose this reflects the interpretation of allometry in an evolutionary context, hence “Evolutionary Allometry”. What about intraspecific variance and moreover, if allometry is being evaluated at this level surely locomotor repertoire should have been evaluated similarly on the basis of species means (this would also allow us to see the position of individual species within Figure 5).

Validity of the findings

The results imply that there is little phylogenetic signal in cranial shape yet the discussion goes on to interpret findings in a phylogenetic context.

Additional comments

Check the referencing to Figures in the results section - e.g. line 223

---

## Round 0.2 · accepted · Accept

· Academic Editor

Accept

Thank you for attending to all the reviewer comments in such a comprehensive fashion. It seemed clear to me that you had addressed all the reviewers' concerns, so I didn't feel it necessary to send the manuscript back out to them. I think this is a really great paper and I look forward to seeing it published.